# In Vivo TSPO Signal and Neuroinflammation in Alzheimer’s Disease

**DOI:** 10.3390/cells9091941

**Published:** 2020-08-21

**Authors:** Benjamin B. Tournier, Stergios Tsartsalis, Kelly Ceyzériat, Valentina Garibotto, Philippe Millet

**Affiliations:** 1Division of Adult Psychiatry, Department of Psychiatry, University Hospitals of Geneva, 1205 Geneva, Switzerland; stergios.tsartsalis@hcuge.ch (S.T.); kelly.ceyzeriat@unige.ch (K.C.); philippe.millet@hcuge.ch (P.M.); 2Department of Psychiatry, University of Geneva, 1211 Geneva, Switzerland; 3Division of Nuclear Medicine and Molecular Imaging, Diagnostic Department, Geneva University and Geneva University Hospitals, 1205 Geneva, Switzerland; valentina.garibotto@hcuge.ch; 4Division of Radiation Oncology, Department of Oncology, University Hospitals of Geneva, 1205 Geneva, Switzerland

**Keywords:** TSPO, Alzheimer’s disease, schizophrenia, astrocytes, microglia

## Abstract

In the last decade, positron emission tomography (PET) and single-photon emission computed tomography (SPECT) in in vivo imaging has attempted to demonstrate the presence of neuroinflammatory reactions by measuring the 18 kDa translocator protein (TSPO) expression in many diseases of the central nervous system. We focus on two pathological conditions for which neuropathological studies have shown the presence of neuroinflammation, which translates in opposite in vivo expression of TSPO. Alzheimer’s disease has been the most widely assessed with more than forty preclinical and clinical studies, showing overall that TSPO is upregulated in this condition, despite differences in the topography of this increase, its time-course and the associated cell types. In the case of schizophrenia, a reduction of TSPO has instead been observed, though the evidence remains scarce and contradictory. This review focuses on the key characteristics of TSPO as a biomarker of neuroinflammation in vivo, namely, on the cellular origin of the variations in its expression, on its possible biological/pathological role and on its variations across disease phases.

## 1. Introduction

The proper functioning of the brain can be disrupted by different types of external factors, such as infection, injury, neurotoxic stimuli and ischemia. In response to these attacks, a dynamic activation of brain cells, especially glial cells, orchestrates a cellular and biochemical response. The set of mechanisms involved forms what is called “acute neuroinflammation.” This process helps eliminate toxic agents, degrades irreversibly damaged tissue, protects surrounding cells and repairs tissue to limit the spread of the damage. Neuroinflammation is, therefore, a physiological process essential to the protection of the central nervous system (CNS). However, the presence of a neuroinflammatory response is not only observed in cases of acute CNS attack. Indeed, a “chronic neuroinflammatory reaction” has been observed in many neurodegenerative diseases, such as Alzheimer’s disease and Parkinson’s disease, as well as in psychiatric diseases such as schizophrenia, depression, and bipolar disorder [1,2]. In these cases, the role and understanding of whether chronic neuroinflammation is a consequence of other pathological processes, or a partner of its etiopathology, is still not fully understood. Consequently, a better understanding of its mechanisms is fundamental to identify the underlying pathophysiological mechanisms.

Evidence for the presence of a chronic inflammatory reaction in the brain can be found through measuring pro- and anti-inflammatory molecules at the central and peripheral level. It has thus been possible to show that in Alzheimer’s disease as in schizophrenia, for example, alteration of the quantities of pro- and anti-inflammatory circulating molecules takes place at the plasma and cerebrospinal fluid levels [3,4,5]. This is also supported by morphological studies within the brain itself. Indeed, *postmortem* studies have made it possible to characterize the neuroinflammatory reaction by the presence of microglia and astrocyte activations [6,7,8,9].

Microglia represents the primary immune effector cell type in the brain and is activated in the presence of injury. From a morphological point of view, microglial cells are relatively small and have many fine extensions. Their morphology is region-specific: Microglial cells in grey matter seem to stretch their extensions in all directions while those with more elongated soma in white matter show an orientation of their extensions along the nerve fibers [10]. The density of microglial cells varies depending on the brain area considered [10,11,12]. Quiescent microglial cells participate in different physiological functions, including the control of synaptic activity [12]. In addition, the extensions act as environmental sensors, measuring the environmental homeostasis, and if necessary, triggering a process of cellular defense. Microglial activation is a complex phenomenon. Indeed, different subclasses of active microglia have been defined based on their gene expression profiles and the type of molecules that they release into the brain parenchyma. Thus, depending on the surrounding environment, the expression of pro-inflammatory proteins can be stimulated. This induces a so-called “M1” profile in microglia [13]. Microglia can also have a rather anti-inflammatory profile, called “M2”, in response to other environmental factors. M1 responses may represent the extreme class mostly associated with cell death, while M2 responses may be related to tissue repair effects [13,14]. In reality, microglial activation probably appears more like a continuum than a dichotomous phenomenon [15]. Other classifications have since been proposed with the identification of clusters of genes making it possible to identify and characterize different subtypes of microglia, such as disease-associated microglia (DAM), neurodegenerative phenotype microglia, and the specific human Alzheimer’s microglial profile (HAM) [16,17,18,19]. The activation of microglia is also under the control of the surrounding cells. For example, neurons can inhibit microglial inflammatory responses via the release of specific peptides whose receptors are localized on microglia [14].

Astrocytes are the second main actor of neuroinflammation. At baseline, astrocytes are pleomorphic and display a set of morphological and functional differences: Protoplasmic or fibrous, depending on their location within the grey or white matter and as a function of the cortical layers of grey matter [20]. The homeostatic regulatory capacity of astrocytes may be elevated, considering that an individual astrocyte can associate with around 100,000 different synapses in the hippocampus [21]. Astrocytes form an interconnected network that releases gliotransmitters (including glutamate), and therefore, participate in the control of neuronal activity [22,23]. They also serve as a metabolic relay between endothelial cells and neurons [24]. Similarly to microglia, astrocytes may exhibit more pro-inflammatory (A1 subtype) or anti-inflammatory (A2 subtype) profiles in response to surrounding factors [25,26], although, as is the case for microglia, the activation pattern of astrocytes is probably best described as a continuum between this A1/A2 polarity [27,28]: The impact of their activity on the tissue, therefore, depends on the balance between pro and anti-inflammatory activation states. It is also interesting to note that other subtypes of astrocytes exist, such as the senescent astrocytes, which also intervene in the neuroinflammatory reaction [29]. In addition, astrocytes emit chemical signals that modulate the activity of microglia, and vice versa [30,31]. For example, in the context of studies on Alzheimer’s disease, it has been shown in vitro that the presence of β-amyloid (Aβ), one of the major players of the pathology, in the culture medium induces an alteration of the microglia’s morphology, into an amoeboid form, characteristic of its activation. This alteration appears to be dependent on the presence of astrocytic cells [32]. In return, microglia cells are able to modify the activation state of astrocytes [26]. A schematic representation of the interactions between neurons, microglia and astrocytes under physiological conditions and in case of neuroinflammation is given in Figure 1.

In the context of Alzheimer’s disease, it is known that the presence of Aβ induces activation of microglial cells and astrocytes [33,34]. Once activated, these cells can take up the soluble Aβ in order to degrade it and eliminate it from the brain [35]. However, it has also been shown that the Aβ loading of microglia and astrocytes can lead to their death by apoptosis. Thus, dead astrocytes and microglia will release the phagocytosed but non-degraded amyloid in the extracellular space. This results in an increase in the size of extracellular amyloid plaques, composed (in large part) of insoluble Aβ [12,14,36,37,38]. This effect has been shown for both white and grey matter astrocytes in animals and humans [36,37]. The presence of amyloid may also participate in neuronal death by conferring a neurotoxic profile to astrocytes [39,40]. The net effect (repairing or deleterious) at the tissue level of the activation states of microglia and astrocytes could depend on the stage of the pathology, the cellular environment or the duration and intensity of the inflammatory reaction [41]. In fact, a low-level inflammation that could be beneficial in the short term may, should it persist, have negative effects, resulting in an inhibition of synaptic functioning that could lead to cell death [12,24]. *Postmortem* analysis of the molecules released by glia made it possible to highlight an upregulation in the presence of cytokines and chemokines favoring a pro-inflammatory environment in neurodegenerative diseases, as well as in schizophrenia [42]. In addition, some astrocytes show morphological signs of atrophy, implying a loss of function in neurodegenerative diseases, as well as in various psychiatric disorders [24,36]. Other morphological effects have also been reported: A hypertrophy of astrocytes with a reduction in their arborization, and a proliferation of microglia with an increase in the size of their soma and a decrease in the size of their extensions. 

In addition to *postmortem* analyses, which among other things have retrospective limitations, many research groups have sought to measure brain inflammation in vivo. In this context, one target appeared to be particularly interesting: The 18 kDa translocator protein (TSPO). Indeed, TSPO is among the very few proteins in the CNS, which is altered in neuroinflammation and which can be studied using in vivo molecular imaging. TSPO was first identified as the “peripheral benzodiazepine receptor” in several organs, such as the heart, blood, gastrointestinal tract, adrenal glands, and liver [2,43,44]. In the CNS, the presence of TSPO in microglial cells is consistently confirmed throughout the literature (Figure 2A). In contrast, its presence in other brain cells is subject to debate. At the cellular level, TSPO displays five transmembrane domains located at sites of contact between the inner and the outer membrane of mitochondria (Figure 2B). A cholesterol-binding amino acid consensus pattern is localized in its cytosolic part and is responsible for cholesterol uptake [45,46]. TSPO takes the form of an 18 kDa monomer, but also of homodimers and homomultimeric forms (Figure 2B) [47]. The multimeric structure is modified by various factors, such as cholesterol concentration and reactive oxygen species; however, the consequences of these modifications are not entirely known [47,48,49]. Thus, TSPO plays a role in transporting cholesterol to the mitochondrial intermembrane space where it is metabolized for steroid formation [50], and in turn can modulate, to a certain extent, neuronal activity [51]. However, its role in the control of steroid synthesis is uncertain [50]. TSPO also appears to associate with other mitochondrial membrane proteins to play a role in controlling the activity of the mitochondrial permeability transition pore (MPTP), and thus, in the mechanisms of apoptosis [52]. The reality of TSPO-MPTP multimeric complex formation does not have a consensus [53,54], but the addition of TSPO ligands in culture medium does cause cell death by apoptosis [55,56,57,58]. 

In vivo imaging of TSPO represents an important challenge in the investigation of brain pathology. More than 50 positron emission tomography (PET) and single-photon emission computed tomography (SPECT) ligands have been developed [59]. In fine, it will help temporally locate the neuroinflammatory reaction regarding the pathology. In addition, the use of TSPO as a biomarker could be important for the development of more effective diagnostic and therapeutic strategies. In the case of Alzheimer’s disease, for example, more than forty preclinical and clinical in vivo imaging studies have been carried out. Importantly, human genetic studies have demonstrated the existence of rs6971 single nucleotide polymorphism (SNP) for TSPO [60] that generates different radioligand binding affinities. Thus, individuals can be classified according to this polymorphism in high-affinity binders (HAB showing the wild-type form of the SNP), low-affinity binders (LAB, rs6971 SNP homozygous) and mixed affinity binders (MAB, rs6971 SNP heterozygous) (Figure 2C). The distribution of the three genotypes in the Caucasian population is at 49% HAB, 42% MAB and 9% LAB [61]. This polymorphism is even less prevalent in Asian countries [28].

Here we review the progress in understanding the role of TSPO as a biomarker in brain diseases. The first part will describe the detection of TSPO levels by in vivo or in situ imaging. Alzheimer’s disease is the most studied pathology in terms of changes in TSPO, in humans and in animal models. Thus, this pathology will serve as a canvas for this review. Among other pathologies, studies on schizophrenia which demonstrate different alterations in TSPO levels from those observed in Alzheimer’s disease will also be described in detail. The second part will discuss the cellular origin of TSPO in different models of chronic and acute neuroinflammation. In the last part, Alzheimer’s disease will be used as an example to address the physiological and pathological role of TSPO. In this section, the links between TSPO, the molecular actors of Alzheimer’s disease (amyloid and Tau) and clinical symptoms will be described. Then, the possible physiological role of TSPO will be discussed.

## 2. TSPO: An In Vivo Marker of Neuroinflammation

### 2.1. Evidence of TSPO Upregulation in Alzheimer’s Disease

Alzheimer’s disease is one of the pathologies with the largest number of studies reporting the use of in vivo imaging of TSPO. Thus, Alzheimer’s disease is the pathology of choice to compare the results obtained between different studies: Is TSPO modified? And if so, in which regions? In addition, as some authors have compared the TSPO results they obtained with different methods of data processing (from the same raw data), it is possible to highlight the bias induced by choice of method. Key data from in vivo PET and SPECT TSPO imaging studies in humans, including clinical and methodological parameters, are given in Table 1.

#### 2.1.1. TSPO Alterations Are Brain Region Specific

Some of the clinical studies report the absence of modifications of TSPO expression related to the pathology [62,63,64,65,66]. This lack of effect was observed both in patients treated with an acetylcholinesterase inhibitors and patients that were not treated [64,65]. A positive effect of age on TSPO levels was also observed [64]. Another study, with significantly older patients (73.7-year-old, range: 60–80) than controls (64.5-year-old, range: 60–73) also indicated a lack of difference between groups [66]. In the Schuitemaker study, the authors also found that TSPO levels were comparable between prodromal Alzheimer’s disease patients who progressed to dementia and those who remained stable during the follow-up period (2.7 years, range: 2.3–3.3) [63]. 

In contrast to these data, most clinical studies reported an increase in TSPO in Alzheimer’s disease [67,68,69,70,73,74,75,77]. However, even though all these studies show an increase in TSPO, there are significant differences in the patterns of its overexpression.

For example, in the hippocampus, one of the regions most involved in Alzheimer’s disease, there is no consensus on the presence of TSPO overexpression. In fact, some studies did not report TSPO alterations in the hippocampus at all [68,69,77]. In contrast, other clinical studies reported an increase in TSPO in the hippocampus in Alzheimer’s disease [70,74,75,76], as well as in mild cognitive impairment (MCI) patients [70,72,74,75,76]. These kinds of differences between studies are not limited to the hippocampus, but are also present in other regions such as the thalamus or the cerebellum [68,69,73]. In order to add further uncertainty to the conclusions of these studies, the interpretation of the TSPO signal within the same studies seems to depend on the methodology applied for its quantification.

#### 2.1.2. Impact of the Method of In Vivo TSPO Quantification

The acquisition of 3D images of the distribution of a radiotracer in the brain is only the prerequisite for understanding the radioactive signal. In fact, in the tissue, all the ligands can exist in different forms: Specifically fixed on their target, free in the parenchyma, non-specifically bound or present in the blood vessels. Thus, post-acquisition image processing is necessary in order to isolate the ligand-specific binding moiety on TSPO. Different methods of signal processing exist and may impact the results of TSPO quantification.

##### Methods of In Vivo TSPO Quantification

Comprehensive analysis of in vivo PET and SPECT imaging relies on the application of analytical methods to the radioactive signal detected by the PET/SPECT camera. In the case where a complete quantification of the binding of a radiotracer on its target is desired, invasive protocols (including one or more injections of radioligands, arterial blood samples, and the dynamic acquisition of the images, e.g., multiple 1-min frames), are necessary. In most cases, the measurement of a binding index whose value is directly related to the total binding (i.e., a semi-quantitative approach) is sufficient to compare the binding between experimental groups and reduces the complexity of the PET/SPECT protocols.

Thus, it is possible to estimate the density of TSPO by calculating the radioligand distribution volume (Vt) that takes into account the number of radioligands in the plasma and the presence of metabolites. The Vt refers to the relationship between the amount of TSPO radioligand in a defined brain area and its concentration in the plasma at steady state. The Vt has been employed as an index of TSPO binding in multiple studies in Alzheimer’s disease [65,66,73,74,75,76]. 

In a less invasive way (i.e., without arterial blood sampling), the calculation of a non-displaceable binding potential (BP_ND_) can be performed. In that case, the binding potential refers to the comparison between the radioligand concentration in TSPO-rich regions and a reference region (where no measurable TSPO binding is present) or pseudo-reference (in which there is some non-negligible TSPO binding, but it is stable across the experimental groups). Estimations of BP were also used to measure TSPO in Alzheimer’s disease [62,63,64,66,68,69,70,71,72]. 

In an even more simplified approach, the binding of TSPO may be quantified on a static image representing the accumulation of radioactivity over a given period of time. The ratio between the radioactive signal in a target-region and a reference/pseudo-reference region can be calculated and employed as an index of TSPO binding. The ratios are calculated from the concentration of accumulated radioactivity in the tissue over a certain timeframe which may be more or less distant from the moment of ligand injection (i.e., the delayed activity). The results are then expressed in standardized uptake ratio (SUR) or standardized uptake value ratio (SUV_R_) depending on the integration or not of the patient’s weight in the mathematical formula. This approach has also been used in various publications related to TSPO in Alzheimer’s disease [62,64,67,75,76,77,78].

However, all the above mentioned simplified methods must first be validated by a more complex method, (e.g., References [79,80,81]). This comparative analysis is crucial to confirm the existence of a reference region and establish the parameters for the acquisition and analysis of images. As has been shown in the literature for other targets than TSPO, failure to validate simplified methods can lead to a discrepancy in results between studies [82]. However, in the case of TSPO, the validation of the various methods to quantify TSPO is subject to debate [83,84,85]. A more detailed description of the different approaches to analyze PET and SPECT exams can be found elsewhere [86,87,88].

##### Impact of the Method of In Vivo TSPO Quantification

As there is still no consensus on how to analyze TSPO images, some authors compared different quantification approaches using the same raw data. Such an approach highlighted non-negligible differences in the results of the quantification depending on the methodological approach. Thus, whilst a region-wise quantification of TSPO showed an upregulation of the protein in the hippocampus in both Alzheimer’s disease and MCI patients compared to controls, a voxel-wise analysis of the exact same images limited this upregulation to the lateral part of the hippocampus and only in the Alzheimer’s disease patients [70]. Similarly, in another set of TSPO PET images, the choice of the quantification approach greatly impacted the statistical significance of the TSPO alteration in Alzheimer’s disease [76]. The interpretation of the TSPO signal in the entorhinal cortex (one of the regions involved early in the course of Alzheimer’s disease) also seems to depend on this method of quantification [75] (see details in Table 1). These comparisons underline the lack of evidence or consensus on which method is the most reliable for quantifying TSPO in vivo. Further studies are, therefore, still needed to shed light on this uncertainty (see also Section 3.2).

#### 2.1.3. Other Influencing Factors

The choice of radiotracer can also affect the quantification of TSPO. Indeed, the [^11^C]PK11195 seems less efficient than the second-generation radiotracers in terms of signal/noise ratio. In addition, [^11^C]PK11195 did not appear as sensitive to the rs6971 polymorphism in contrast to all the other radiotracers used to measure TSPO. Thus, studies using second-generation radiotracers that do not consider this polymorphism may have introduced a bias in their measurement. However, although the results obtained are sensitive to this polymorphism, in the case of Alzheimer’s disease, the results seem to converge in the sense that both HAB-AD and MAB-AD show higher levels than their respective controls [73,74,77]. It is difficult to conclude for LAB-AD, due to the small number of patients studied (Table 1).

The timing of scan acquisition can also impact the result. Indeed, when using the SUV_R_ as a density index of TSPO, it seems highly probable that the ratios will be different when the images are acquired between 0 and 60 min post-injection [68,69] vs. late images acquired between 60 and 80 min post-injection [67] using [^11^C]PK11195. Finally, the very use of the cerebellum in the SUV_R_ method as a (pseudo-)reference region [62,64,66,67,75,76,77] could be questioned given the presence of an upregulation of TSPO in this region between Alzheimer’s disease patients and healthy control subjects [68,69,73]. 

The selection of patients may also have an impact on the results. In fact, while age probably influences the density of TSPO [89] and gender influences neuropathological and clinical progressions of Alzheimer’s disease [90], in some studies, participants in the patients’ and the control group were not age- or sex-matched (see details in Table 1). 

A final limitation of clinical studies of TSPO in Alzheimer’s disease is the number of participants: Ranging from 6 to 20 across studies. The problem of sample size becomes even more important when TSPO polymorphism status is considered. In a study where the presence of TSPO is evaluated according to the TSPO genotype, this results in a decrease in the number of patients per sub-group. For example, in the healthy HAB group positive for amyloid (HAB-HC-A^+^), there were only two individuals (see Reference [77] in Table 1).

However, a recent meta-analysis, including 269 AD patients, 168 MCI. And 318 healthy controls, confirmed the presence of an increase in TSPO in Alzheimer’s disease [91].

### 2.2. Evidence of a TSPO Upregulation in Preclinical Models of Alzheimer’s Disease

#### 2.2.1. TSPO Is Increased in Tau, APP/PS1 and APP/PS1/Tau Mouse Models

In humans, although efforts still need to be made to better characterize the presence of TSPO, it is now widely accepted that it is overexpressed in brain regions in Alzheimer’s disease patients [91]. Several studies have also sought to show the existence of overexpression of TSPO in different animal models of the pathology.

An age-dependent increase in TSPO levels in the hippocampus was observed in both Tau transgenic (Tg) [92,93], APP Tg [92,93,94,95], and double (APP/PS1) Tg [96,97,98,99,100] mice. In our recent study in triple (APP/PS1/Tau) Tg model (3xTgAD), we demonstrated a subregion-dependent effect of age on TSPO levels in the hippocampus [101]. Indeed, TSPO is upregulated at 6 months in the subiculum, at 12 months in the dorsal and anterodorsal hippocampus and only at 21 months in the ventral hippocampus. These data show that TSPO upregulation within the hippocampus is not homogeneous.

#### 2.2.2. TSPO Binding in Cerebellum and Choroid Plexus

Although most animal studies conclude that TSPO increases in brain regions in Alzheimer’s disease animal models, it is important to note that methodological issues inherent to small animal imaging may have influenced the results. Two tissues involved in the analyses express TSPO and are important to describe in this context: The cerebellum and the choroid plexus (Figure 3).

In some in vivo PET or SPECT imaging studies, an index of TSPO density is calculated by the SUR or SUV_R_ ratios with the cerebellum as a pseudo-reference region (see Section 2.1.2). However, other studies showed that not only the TSPO signal of the cerebellum is specific, but it may possibly be elevated [98,99] in Tg compared to wild-type (WT) animals, though this result has not been reproduced in all the relevant studies [97,100]. In a recent study, the cerebellum was used as a reference region to normalize TSPO binding levels, whereas a significant increase in cerebellar uptake was present in the old WT mice group [103]. Thus, the cerebellum may not represent a good reference region. Other studies have used the striatum, the thalamus, the whole brain or the heart as reference region [93,96,97], but a more in-depth study of the validity of these regions as a reference will be needed.

Another limitation in small animal imaging studies of TSPO may be the high specific signal in the choroid plexus. The choroid plexuses, located in the ventricles, highly express TSPO, both in humans and in laboratory animals [94,101,102]. Thus, choroid plexuses present a considerable TSPO radiotracer binding. Given the anatomic proximity between the ventricles and the hippocampus, we can suggest a high probability of mutual spillover of the radioactive signal between these two regions. In support of this hypothesis, we showed by SPECT with [^125^I] CLINDE in mice that the amount of radioactivity measured between the hippocampus and the choroid plexuses were highly correlated, which suggests an interdependence of the concentration of measured radiotracer concentration in these two regions and suggests that the radioactive signal measured in vivo in the hippocampus may, in part, originate from the nearby choroid plexus [101]. An elevated binding to the choroid plexus and the spillover effect on the hippocampus was also observed ex vivo in small animal studies. Regarding human imaging of TSPO, the large size of the brain compared to rodents suggests that the spillover effect could be of lesser importance than in small animals, but this has yet to be thoroughly studied.

### 2.3. Is TSPO Altered in Other Pathologies? 

The number of studies concerning TSPO in other psychiatric and neurodegenerative pathologies is much less important than in the case of Alzheimer’s disease. Thus, the conclusions are supported by a smaller number of data, and as in Alzheimer’s disease, can be contradictory between studies. 

It can be noted that, overall, studies have shown an upregulation of TSPO in neurodegenerative diseases, such as Parkinson’s disease and Huntington’s disease, as well as in multiple sclerosis [61,83,104]. 

In contrast, schizophrenia represents a very particular state in terms of TSPO expression. Early studies reported a lack of effect, a decrease or an increase in TSPO vs. controls, but these studies not necessarily considered the TSPO polymorphism [2,85,105,106,107,108,109,110,111,112,113,114,115] (see Table 2). Clinical and methodological differences can at least partly explain discrepancies between studies [85,114]. For example, the presence and dose of antipsychotic treatment are not identical between studies. This status can also vary greatly in the same study, as shown by Selvaraj and coll., where one patient was antipsychotic-free, two had an unknown status, and eleven were under treatment [112], or in the Notter study [111], where the ratio of the antipsychotic dose between patients was greater than 1000 (in terms of chlorpromazine equivalent doses). Another important factor: While nicotine is also suspected of inducing changes in TSPO binding, its consumption is not always considered and can even be an important confounding factor between groups, as in the Holmes and coll. report, where schizophrenic patients were smokers while controls were not [109]. 

However, a recent meta-analysis of five studies (75 patients and 77 healthy controls) that considered the TSPO polymorphism shows a decrease in TSPO in schizophrenia [116]. Interestingly, a meta-analysis of *postmortem* brain pathology studies (783 patients and 762 controls) concluded that microglial alterations and overexpression of pro-inflammatory molecules are associated with schizophrenia [42]. Similarly, signs of astrocytic alterations have been reported [117]. Thus, both in schizophrenia and Alzheimer’s disease, microglial and astrocytic alterations are present. Consequently, one should expect that, in both conditions, in vivo imaging studies of TSPO would show an upregulation of TSPO binding. However, this is not the case, as schizophrenia seems to be associated with a decrease in TSPO, while Alzheimer’s disease shows an increase in TSPO.

Finally, in other pathologies, such as bipolar disorder, few studies have examined TSPO—which makes reaching a firm conclusion impossible [118].

## 3. Cell Origin of TSPO Alterations

In the brain, it has historically been considered that the sole origin of TSPO is microglia. This idea has since been largely challenged by evidence of the presence of TSPO in other brain cell types. However, the idea still persists that TSPO is only an indicator of microglia activation.

### 3.1. TSPO in Astrocytes and Microglia 

All *postmortem* studies of cell colocalization performed in humans showed the presence of TSPO in microglia, either through CD68^+^ colocalization with the autoradiographic image of a TSPO radiotracer or through TSPO^+^ colocalization with a microglial marker (Iba1, CD68 or HLA-DR according to studies) [96,97,100,119,120,121]. On the contrary, the colocalization of TSPO^+^ with GFAP^+^ in astrocytes has not consistently been observed [96,97,100,119,120,121].

In animal models, the same conclusions can be drawn: TSPO is present in microglia in various Alzheimer’s transgenic mice (including APP/PS1, 3xTgAD, 5xFAD and Tau mice P301S), but not all of them (APP23 Tg mice). Contradictory results on the presence of TSPO in astrocytes were also reported in these studies. Although some reports described TSPO in astrocytes, others did not [92,96,97,100,101]. In addition, even in the studies that reported a presence of TSPO in astrocytes, staining can be either fairly dense or on the contrary scarce according to the AD model used. Figure 4 presents the results of double-immunostaining experiments in tissue sections from the brain of TgF344-AD rats [122]. 

The expression of TSPO in astrocytes could possibly be region-dependent. Indeed, in the frontal cortex, four out of six studies showed the presence of TSPO in microglia and the absence of astrocytic expression of TSPO [97,100,120,121]. Conversely, all studies in the hippocampus show a double colocalization of TSPO with microglia and astrocytes [92,96,101,119]. 

As described in the introduction, microglia and astrocytes present a considerable phenotypic heterogeneity. However, it should be noted that, to date, only a handful of studies have sought to determine which sub-types of microglia (M1, M2) or astrocytes (A1, A2, protoplasmic or atrophic forms) express TSPO. Among these studies, Liu et al. showed colocalization of TSPO with CD86^+^ cells (a pro-inflammatory marker), but also with CD206^+^ cells (an anti-inflammatory marker) when these cells were located near amyloid plaques [96].

In order to reveal TSPO-M1 or TSPO-M2 associations, several studies have been conducted in vitro on microglial cells. In rodent cell lines, the presence of a pro-inflammatory agent in the medium resulted in the formation of M1-type microglia and overexpression of TSPO. Conversely, the presence of anti-inflammatory agents did not lead to changes in TSPO, either in vitro or after intracerebroventricular injection [123,124]. In human adult microglia primary cell cultures, the addition of pro-inflammatory molecules did not result in changes in TSPO expression but induced a decrease in protein amounts [123]. Even if this observation concerns a small sample size, this implies that TSPO would not be overexpressed in each cell of the microglia but rather decreased. A decrease in TSPO (mRNA and/or protein) has also been observed in monocyte-derived M1-macrophage [123,125]. 

To determine the time course of the appearance of TSPO in astrocytic and microglial cells, a recently validated approach, the fluorescence-activated cell sorting to radioligand-treated tissues (FACS-RTT) was used in TgF344-AD rats at the age of 12 and 24-months. Data showed that TSPO is expressed in both cell types, but its overexpression in astrocytes predated that in the microglia [122]. The precocity of the astrocyte reaction agrees with other studies in animal models of Alzheimer’s disease and with the presence of an astrocytosis from the mild cognitive impairment stage in Alzheimer’s disease patients [30,126,127,128].

### 3.2. Other Cells Types

An immunohistochemical study of the brain of WT mice made it possible to demonstrate the presence of TSPO in the endothelial cells, astrocytes, microglia, pericytes and purkinje cells, as well as a lack of labelling in neurons and oligodendrocytes [102]. However, the cellular origin of TSPO is not homogeneous and depends on the considered region. Thus, in the hippocampus where TSPO is preferentially expressed in the subgranular zone, the following distribution was observed: CD31^+^ cells > GFAP^+^ and Nestin^+^ cells, with an absence of expression by CD11b^+^ microglial cells. TSPO immunoreactivity in CD31^+^ cells indicate an endothelial origin of TSPO, GFAP^+^ can indicate both astrocytes, and neural stem cell origin of TSPO and the Nestin^+^ signal demonstrates a neural stem cell origin of TSPO. By contrast, the cortex did not show positive labelling for either astrocytes or microglia. In rats, positive staining in endothelial cells of the hippocampus was shown (Figure 4). In humans, TSPO is also observed in endothelial cells and in vascular smooth muscle cells [120]. TSPO could also be expressed by neurons [83,120].

The FACS-RTT approach showed that, in human *postmortem* tissue samples, the endothelial cells of the frontal and temporal cortex presence TSPO binding—which is not altered in Alzheimer’s disease [122,129]. On the other hand, we showed that TSPO increases are due to astrocytes and microglial cells (Figure 5). The latter shows an expansion without variation in the number of TSPO binding sites per cell. Interestingly, when we measured the binding of TSPO on the whole tissue (i.e., before cell sorting), we were not able to observe any TSPO increase in Alzheimer’s disease in the frontal cortex [129]. This underlines the importance of looking for effects at the level of the identified cell populations. 

### 3.3. Cells Expressing TSPO in Other Pathologies

The cellular origin of TSPO binding alterations may not be common across brain conditions. Indeed, in schizophrenia where TSPO binding most probably decreases, results of preclinical studies suggest that this decrease concerns microglia, astrocytes and endothelial cells [111]. Due to the involvement of endothelial cells in the downregulation of TSPO, the interpretation of the PET/SPECT imaging data must be made with caution and any TSPO alterations measured in vivo should not be attributed to alterations in glial cells only.

Interestingly, in a cuprizone-induced demyelination model, TSPO appeared first in microglia, followed by astrocytes during the demyelination phase. This upregulation of TSPO persisted in astrocytes after the treatment was stopped [2]. Thus, the involvement of the different cell types in the alterations of TSPO seems to depend on the type of pathology.

## 4. Possible Biological Roles of TSPO

Even if TSPO is widely used in preclinical and clinical PET/SPECT imaging as a marker of neuroinflammation, its role in physiology and pathophysiology is not fully understood. The TSPO protein may have interactions with the two major molecular players in Alzheimer’s pathology (Aβ and Tau), it may be linked to clinical symptoms and may have a protective or an aggravating role in the pathology.

### 4.1. Relationship between TSPO and Alzheimer’s Disease

#### 4.1.1. TSPO and Aβ Relationship

When the distribution of TSPO in the brain in relation to the distribution of amyloid plaques was measured with PET using [^11^C]PK1119/[^18^F]FEPPA and the ^11^C-Pittsburgh compound-B, ^11^C-PiB radiotracer, respectively, it was shown the binding of the two radiotracers were positively correlated in some studies [70,71,72,77]. However, this finding was not confirmed in other studies [62,69]. In the hippocampus of preclinical models, a positive TSPO/Aβ deposits correlation was mainly reported [93,95,101,130]. However, our preclinical data tend to show that the subregions of the hippocampus do not react in the same way. In the 3xTgAD mouse and in the TgF344-AD rats, we showed a positive correlation between TSPO and amyloid plaques in the dorsal hippocampus that is absent in the ventral hippocampus (Figure 6). Thus, the strength of the functional links between TSPO (i.e., the inflammatory reaction) and the presence of amyloid deposits may depend both on the area under consideration and on the progress of the pathology. It is also important to keep in mind that the lesions associated with AD pathology are not uniformly in the patients’ brains [131,132].

In humans, studies have shown that TSPO-related neuroinflammatory reaction is an early phenomenon with respect to tau pathology and amyloid plaques, suggesting that neuroinflammation may not be a late consequence of pathology but rather participates in its evolution [69,72,77]. However, this idea is also controversial [74]. In 3xTg-AD mice, we showed that, in all hippocampal subregions, the upregulation of TSPO occurs before the presence of amyloid deposits as detected by the [^125^I]CLINDE and [^125^I]DRM106 bindings, respectively, supporting the idea of TSPO’s early role in with respect to Alzheimer’s disease associated brain lesions [101].

#### 4.1.2. TSPO and Tau Relationship

The presence of an inflammatory reaction related to the accumulation of pathological forms of Tau has been widely reported [34,133,134,135,136,137,138]. However, few studies have directly analyzed the link between TSPO and Tau. In the P301S Tau Tg mice, a positive correlation was observed [93]; however, this appears to be absent in humans [139]. However, two limitations of the clinical study [139] should be reported: A small number of patients (six Alzheimer’s disease patients at an early stage) and the use of a first-generation TSPO ligand showing suboptimal imaging properties. Thus, further studies are still needed to fully understand the links between TSPO and tauopathy (see also Section 4.2).

#### 4.1.3. TSPO and Clinical Symptoms of Alzheimer’s Disease

The density of TSPO could also be related to the cognitive status of patients, estimated by MMSE and/or the clinical dementia rating (CDR) scores, depending on studies. But while most of the reports showed a negative correlation (i.e., high TSPO levels in association with increased deficits) [67,69,70,74,75], others did not report any links [63,73] or even a positive correlation [77,78].

Supporting the idea of early implication of TSPO in pathology, it has been shown that patients with high initial rates of TSPO have lower cognitive deficits in the following years [77,78,140]. On the contrary, patients with the highest increases in TSPO during this same period showed a higher decrease in their cognitive abilities [76,77,78].

#### 4.1.4. Is TSPO Protector or Aggravator?

Analyses of positive correlations between the amounts of TSPO, amyloid plaques and phospho-Tau on the one hand and cognitive loss on the other hand (see details in Section 4.1) could imply that TSPO plays a rather aggravating role in the pathology. Moreover, patients showing the highest increases in TSPO during a longitudinal follow-up also showed the greatest cognitive loss [78].

At the cellular level, at least some of preclinical studies support this hypothesis. The administration of TSPO antagonists in animals induced a decrease in Aβ levels in the hippocampus [141] and reduced inflammatory effects in response to an injection of pro-inflammatory lipopolysaccharide (LPS) [142,143,144,145]. Conversely, lentiviral-mediated TSPO overexpression decreases the cognitive deficits associated with LPS [146,147]. However, a significant limitation must be considered when interpreting these results. Indeed, the use of lentivirus very probably introduces a strong overexpression of TSPO in neurons that certainly exceeds that of microglia because of their cell density. As these studies did not control for the cell type causing the signal increase, the effect cannot only be considered to be glial-mediated. Moreover, as described in the previous paragraphs, neuronal TSPO seems to play little or no role in neurodegenerative diseases.

Currently, no data is relevant to the role of TSPO in astrocytes in terms of protective or deleterious function. On the other hand, changes in TSPO (increases/decreases) in microglia appear to only affect the M1 subtype [124,125], which also supports the hypothesis of a rather pro-inflammatory function. However, it cannot be totally ruled out that the increases in TSPO make it possible to limit and not facilitate the M1 type function. Future studies are therefore needed to clarify the role that TSPO plays in microglial and astrocytic responses.

### 4.2. Relationship between TSPO and Schizophrenia

In schizophrenia, a positive correlation between TSPO levels and negative symptoms has been reported [108,109] and may suggest functional links between the presence of inflammation and the onset of negative symptoms. Indirect evidence supports this idea, such as the exacerbation of some negative symptoms in response to pro-inflammatory peripheral stimulation [148,149,150,151,152,153,154]. In addition, a meta-analysis showed an improvement of negative symptoms with the consumption of the antibiotic minocycline, which possesses anti-inflammatory properties [149].

## 5. Conclusions and Perspectives

Overall, the literature agrees that TSPO can be either upregulated or downregulated according to the pathology considered. The cellular origin of TSPO alterations are always at least microglial, but may also affect astrocytes and endothelial cells, which complicates the interpretation of TSPO as a marker of glial activity. From a biological point of view, evidence of a protective effect, as well as indications of an aggravating effect of the pathology, has been observed. It is thus currently difficult to reach a conclusion on this matter, and we cannot exclude the presence of a dual effect of TSPO depending on the degree of activation/overexpression or the stage of the pathology.

Beyond cell origin, cell culture approaches seem to show a preferentially M1 phenotype, but it is important to note that the in vitro and in vivo physiology of microglial cells is very different [14]. Future studies should make it possible to specify which sets of cells modify their TSPO expression in terms of activity. Indeed, in vivo imaging approaches cannot consider the diversity of microglia and astrocytes. The A1/A2 and M1/M2 segregation is, of course, a simplified view of all the forms that these cells can take [28]. To further complicate matters, certain forms of microglial cells seem to appear as the pathologies evolve. This is the case, for example, of the existence of the dark microglia subpopulation [155] or the protective disease-associated microglia (DAM) [17]. Interestingly, TSPO was also reported in a microglial subtype clustered with LPS-associated genes [156]. However, it would be interesting to study in which specific microglial subtype the TSPO is modified. This diversity of glial phenotypes (themselves impacted by the degree of pathology) could perhaps explain, at least in part, the differences between pathologies concerning TSPO expression, as well as the differences between studies.

It is also possible that different regions of the brain do not overexpress TSPO in the same way (as seems to be the case in a physiological situation, see Section 3.2). Indeed, the variety of forms of astrocytes and microglia as a function of their location, variations in the density of microglia between tissues, the presence or absence of TSPO in a particular cell type at rest and the cellular and chemical environment could be the origin of a region-specific response. For example, correlations between TSPO and amyloid are observed in the hippocampus but not in the cortex, and to put it simply, TSPO may be expressed in the astrocytes of the hippocampus but not in those of the cortex. Future studies will, therefore, have to answer the question of the presence of tissue sensitivity. Similarly, it seems important to better characterize TSPO according to the activity state of the glia, but also according to its location: In other words, do the grey matter astrocytes and microglia, whose phenotypes are different from those in white matter, react in the same way in terms of TSPO expression? Finally, future studies will need to consider whether radioligand binding to TSPO is influenced by homomultimerization. Indeed, some studies report a variation of this capacity in culture models without alteration of total TSPO density [157].

The constraints and uncertainties brought about by in vivo imaging of TSPO can lead to this fundamental question: Should we continue TSPO investigations? It is clear that many uncertainties remain, either for the methods of data quantification, the methods of patient selection (exclusion in all the latest studies of 10% of the population characterized as LAB) and the interpretation of the signal between pathologies (Alzheimer’s disease vs. schizophrenia). However, TSPO imaging is still important in at least two fields of application. The first is to provide diagnostic assistance to the early identification of pathologies in order to implement therapeutic strategies as soon as possible. In this respect, it seems that TSPO is modified early in Alzheimer’s disease and in schizophrenia, although other studies have yet to confirm this idea (see Table 1 and Table 2). The second field of interest will be to use TSPO as a marker of treatment effectiveness. For example, cognitive improvement (or cessation of aggravation) may be compared to changes in TSPO binding.

Apart from using TSPO as a tool for measuring neuroinflammation in vivo, a better understanding of its physiological and pathological functions could make TSPO a target for therapeutic action. In this view, several preclinical studies have shown that the use of TSPO agonists or antagonists may have a protective effect in the context of Alzheimer’s disease [141,158,159]. This understanding is particularly important, given the lack of efficient treatment strategies [160].

Thus, future studies are still needed to clarify the role of TSPO as a marker in the presence of pathology, as well as a marker of its evolution and target of future therapeutic agents.

## Figures and Tables

**Figure 1 cells-09-01941-f001:**
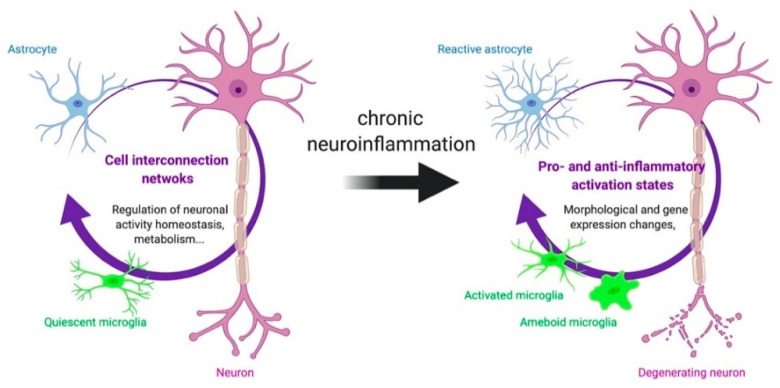
Interactions between neurons, astrocytes and microglia in health and disease. In a healthy brain, the connectivity between neurons, astrocytes and microglia maintains homeostasis. In brain disease, specific morphological and functional alterations of glial cells could lead to brain degeneration.

**Figure 2 cells-09-01941-f002:**
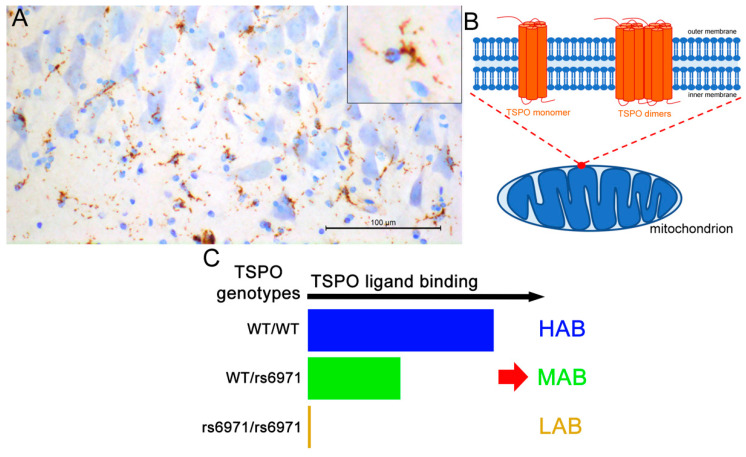
Overview of the 18 kDa translocator protein (TSPO). (**A**) TSPO-immunoreactivity in the human CA1 subregion of the hippocampus. Staining looks like microglial morphology. (**B**) Schematic representation of TSPO monomers and dimers within the mitochondria. Homomultimeric forms exist (not shown). (**C**) Impact of the human TSPO rs6971 polymorphism on TSPO ligand binding and resulting population classification in HAB (high-affinity binder), MAB (mix affinity binder) and LAB (low-affinity binder). Scale bar: 100 μm.

**Figure 3 cells-09-01941-f003:**
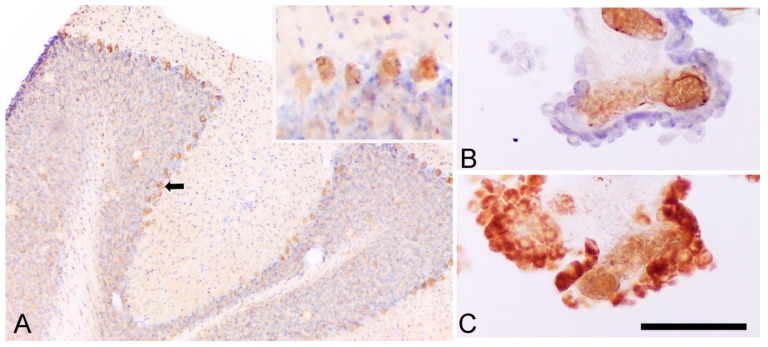
TSPO is expressed by Purkinje and choroid plexus cells. (**A**) The immunoreactivity of rat cerebellum to TSPO (with cresyl violet counterstaining) underlines a positive staining in Purkinje cells (the black arrow indicates an example and insert), as demonstrated in mouse cerebellum [102]. (**B**) Immunoreactivity of human choroid plexus cells to CD31 (endothelial cells) with cresyl violet counterstaining (epithelial cells). (**C**) Immunoreactivity of human choroid plexus cells to TSPO underline both epithelial and endothelial cells positive for TSPO (as previously published in Reference [101]). Scale bar: 250 μm.

**Figure 4 cells-09-01941-f004:**
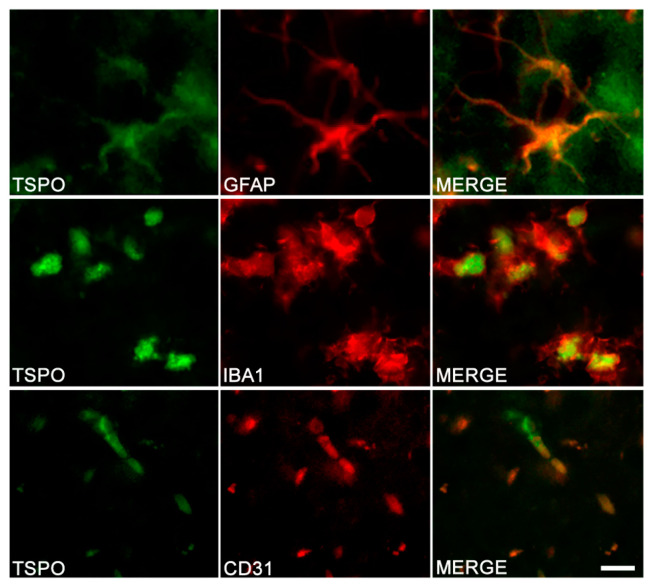
TSPO in different cell types of the hippocampus in TgF344-AD rats. Double-immunostaining was performed to detect TSPO (left column) and specific marker of astrocytes (GFAP), microglia (IBA1) and endothelial cells (CD31). Merge images demonstrate the colocalization of TSPO with astrocytes, microglia and endothelial cells. Scale bar: 10 μm. Adapted from Reference [122].

**Figure 5 cells-09-01941-f005:**
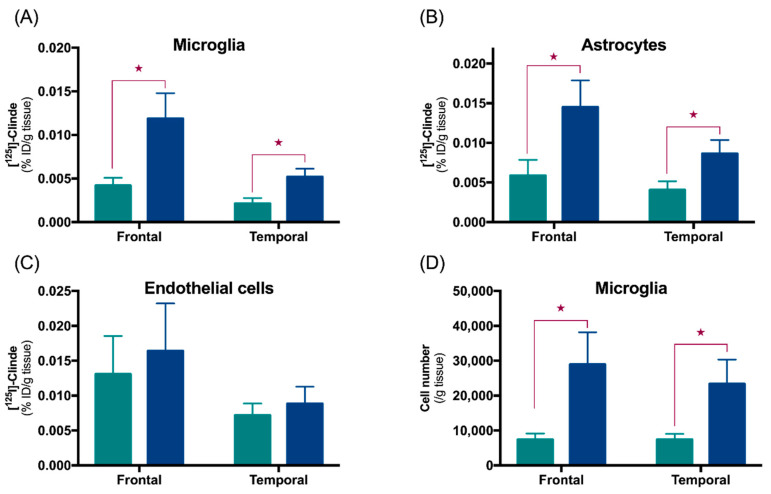
TSPO is increased in astrocytes and microglial cells in the frontal and temporal cortex of Alzheimer’s disease subjects. (**A**–**C**) The [^125^I]CLINDE binding is used to assess TSPO density in the frontal and the temporal cortex of control (green) and AD (blue) samples. Radioactivity was determined in microglia (CD45^+^ cells), astrocytes (GLT1^+^ cells) and endothelial cells (CD31^+^ cells) and expressed as % injected dose (ID) /g of tissue. (**D**) The number of microglial cells sorted. ★: *p* < 0.05. Adapted from References [122,129].

**Figure 6 cells-09-01941-f006:**
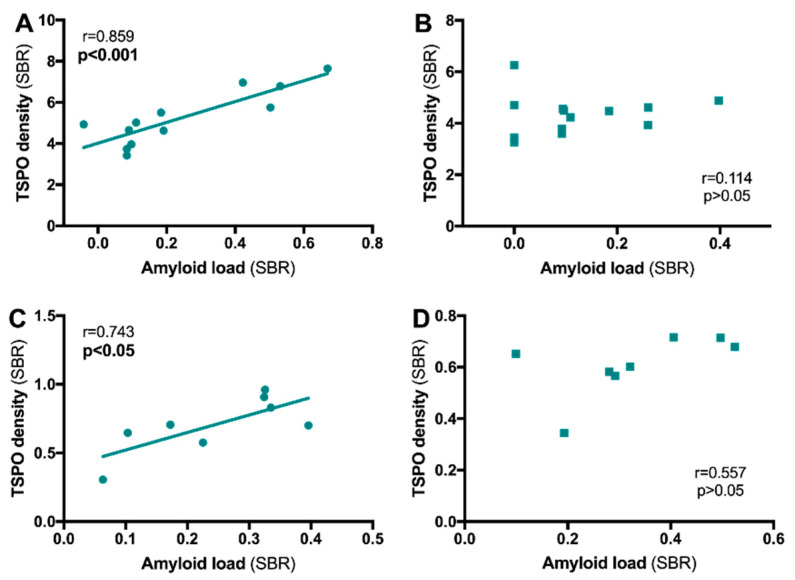
Relationship between TSPO density and amyloid load in the hippocampus of 3xTg-AD mice and TgF344-AD rats. TSPO density and amyloid load were determined by in situ autoradiography with [^125^I]CLINDE and [^125^I]DRM106 in 3xTg-AD mice (**A**,**B**) and TgF344-AD rats (**C**,**D**). A positive correlation was observed in the dorsal hippocampus (**A**,**C**) but not in the ventral hippocampus (**B**,**D**). r = Spearman’s correlation coefficient. Mouse data were published in Reference [101], and rat data are personal observations.

**Table 1 cells-09-01941-t001:** Main results of TSPO positron emission tomography (PET)/single-photon emission computed tomography (SPECT) studies in Alzheimer’s disease.

Radiotracer	Population	m/f (%males)	Age Range (mean)	MMSE Score Range (mean)	TSPO Methodology: Scan Acquisition and Analysis	Main Results	Reference
***Studies showing no main effect of AD on TSPO***							
^11^C-PK11195 * ^11^C-PiB	6 modAD-A^+^ 6 MCI (4 AD-A^+^) **5 HC** (2 HC-A^+^)	4/2 (67) 4/2 (67) 3/2 (60)	65–94 (76) 61–81 (72) 65–79 (72)	13–28 (19.3) 27–30 (28.7) 28–30 (29.4)	0–90min BP_SRTM_ with Ref_CE_ 10–60min images SUR with Ref_subcortical white matter_	BP_SRTM_ and SUR: **AD = MCI = HC** **A^+^ = A^−^**	[62]
^11^C-PK11195 * No amyloid detection	20 AD 13 proAD 21 HC	11/8 (58) 7/3 (70) 13/8 (62)	(69) (72) (68)	(23) (26) (29)	0–60.5 min. BP_SRTM_ with Ref_Cluster Analysis_ VWA-SPM	BP_SRTM_: **AD = prodromal = HC** VWA-SPM: **AD > HC**:Occ No correlations between PK11195 and neuropsychological tests. No difference between prodromal AD patients who remained clinically stable and those who progressed clinically to dementia.	[63]
^11^C-Vinpocetine No amyloid detection	6 AD 6 young HC 6 old HC	3/3 (50) 6/0 (100) 6/0 (100)	67–82 (73) 54–78 (67) 25–44 (35)	(dnf)	0–66 min SUV_R_ with Ref_CE_ BP_Logan_ with Ref_CE_	SUV_R_ **Old HC > Young HC**: pF, medT, latT, Occ, Th, St **AD = Old HC** BP_Logan_ **AD = HC**: Th **AD = Old HC > Young HC**: whole-brain, grey matter	[64]
^18^F-FEDAA1106 No amyloid detection	9 AD 7 HC	6/3 (67) 5/2 (71)	64–76 (69) 63–73 (68)	21–30 (25) 28–30 (29)	0–60 min and 80–140 min. V_T_ BP_ND_ V_T_ Logan	AD: 8 AChE inhibitor + 1 AchE inhibitor- V_T,_ BP_ND_, V_T_ Logan **AD = HC**	[65]
^18^F-DAP-714 No amyloid detection	9 AD 6 HC	4/4 (50) 1/5 (17)	60–80 (73.7) 60–73 (64.5)	20–29 (24.5) 28–30 (28.8)	0–90 min and 120–150 min V_T_ BP_SRTM_ with Ref_CE_	**Age effect: AD > HC** V_T_, BP_SRTM_ **AD = HC**	[66]
***Studies showing TSPO overexpression in AD***							
^123^I-PK11195 * No amyloid detection	10 AD 9 HC	**4/6 (40)** **6/3 (67)**	55–87 (77) 53–76 (67)	9–25 (19)	60–80 min SUR with Ref_CE_	**HC**: no age effect. **AD > HC**: loF, pF, latF, rMT, bG Diverse inverse correlations between regional PK1195 and neuropsychological tests.	[67]
^11^C-PK11195 * No amyloid detection	8 AD 15 HC	4/4 (50) 7/8 (47)	58–68 (65) **32–80 (75)**	6–24 (17.25)	0–60 min. BP_SRTM_ with Ref_Cluster Analysis_	**HC**: age effect in Th **AD > HC**: iTG, mTG, fG, lPG, lAm, lpCin, iPG, Pu, rPall **AD trend to > HC**: rCE (p=0.06)	[68]
^11^C-PK11195 * No amyloid detection	13 AD 10 HC	8/5 (62) 6/4 (60)	54–73 (65) 54–71 (64)	15–26 (21) (30)	0–60 min. BP_SRTM_ with Ref_Cluster Analysis_ VWA-SPM	BP_SRTM_ **AD = HC**: Th, Hipp **AD > HC**: aCin, pCin, St, F, T, P, Occ, WC, Am, PG, mTG, CE (+22%) VWA-SPM: **AD > HC**: iTG, mTG, mFG, rF, latOcc, loF, lpostCG, rpostT, infF, roF, preCG, supFG. Negative correlations between regional PK11195 and MMSE scores	[69]
^11^C-PK11195 * ^11^C-PiB	10 AD-A^+^ 10 MCI (5 MCI-A^+^) 8 HC	4/6 (40) 5/5 (50) 4/4 (50)	51–74 (66) 55–77 (67) 58–71 (65)	(20.5) (28.2) (30)	(dnf) min BP_SRTM_ with Ref_Cluster Analysis_ VWA-SPM	BP_SRTM_ **AD > HC**: aCin, F, T, P, Occ, mTG, Am, Hipp **MCI > HC**: aCin, F, T, P, Occ, mTG, Am, Hipp, pCin VWA-SPM **AD > HC**: lHipp, rPG, lpostT, lpreCG, lpostCG, lsupTG, liTG, lmTG, latOcc. **MCI > HC**: liTG, lmTG, rsupFG, rsupPG, lIns, lPu, rmedOG, lmFG, raOG, rStG, rlatOcc In different regions (AD only): Positive correlations between PK1195 and MMSE scores; Positive correlations between PK1195 and PiB	[70]
^11^C-PK11195 * ^11^C-PiB	8 AD (7 AD-A^+^) 8 HC out of 14	3/5 (37) 7/9 (44) **	51–74 (66) 54–75 (65) **	(21)	0–60 min BP_SRTM_ with Ref_Cluster Analysis_ VWA-SPM	BP_SRTM_ and VWA-SPM **AD > HC**: F, T, P, Occ, Hipp, St Positive correlations between PK1195 and PiB	[71]
^11^C-PK11195 * ^11^C-PiB	26 MCI-A^+^ 16 MCI-A^−^ 10 HC out of 15	17/9 (65) 7/9 (44) 6/) (40) **	62–83 (73) 50–79 (66) 58–80 (68) **	23–30 (27) 23–30 (28) 25–30 (29) **	0–60min BP_SRTM_ with Ref_Cluster Analysis_	**MCI-A^+^ > HC**: F, latT, P **MCI-A^+^ > MCI-A^−^**: Hipp **MCI-A^−^ = HC** In different regions: positive correlations PK1195 and PiB	[72]
^18^F-FEMPA No amyloid detection	5 HAB-AD 3 MAB-AD 2 ?-AD 4 HAB-HC 3 MAB-HC	2/3 (40) 2/1 (67) 1/1 (50) 1/3 (25) 2/1 (67)	67–73 (71.2) 55–67 (61) 56–74 (65) 66–71 (69) 55–58 (56)	23–28 (25.6) 22–28 (25) 23–29 (26) 28–30 (29) 29–30 (29.7)	0–90 min and 120–150 min. V_T_ V_T_ Logan	V_T_ Logan **HAB/MAB-AD > HAB/MAB-HC**: medT **HAB-AD > HAB-HC**: medT, latT, pCin, Cau, Pu, Th, CE (+19%) No correlations between FEMPA and MMSE scores	[73]
^11^C-PBR28 No amyloid detection	9 HAB-AD 10 MAB-AD 4 HAB-MCI 6 MAB-MCI 5 HAB-HC 8 MAB-HC	11/8 6/4 9/4	(63.1) **(72.6)** (62.9)	(20.3) (27.5) (29.8)	0–90 min V_T_	**Age effect: MCI > AD = HC** AD: 15 AchE inhibitor +, 4 AchE inhibitor- MCI: 4 AchE inhibitor +, 6 AchE inhibitor- **HAB > MAB** (controls and patients combined) in whole brain. **AD > HC**: pF, iP, supT, iTG, mTG, Prec, pCin, Occ, Hipp, Ent **AD > MCI**: pF, iP, supT, iTG, mTG, Occ **HAB-AD > HAB-HC**: iP **HAB-AD > HAB-MCI**: iP **MAB-AD > MAB-HC**: iP Correlation between regional PBR28 (AD and MCI) and neuropsychological tests, grey matter volume and age of symptom onset.	[74]
^11^C-PBR28 ^11^C-PiB	11 HAB-AD-A^+^ 14 MAB-AD-A^+^ 5 HAB-MCI-A^+^ 6 MAB-MCI-A^+^ 7 HAB-HC 14 MAB-HC	AD: 11/14 MCI: 7/4 HC: 15/6	AD: (63) MCI: (72) HC: (55)		0–90 min. V_T_ SUV_R_ with Ref_CE_ (60–90 min) DVR (V_T_ target/V_T_ CE)	***Some subjects were previously included in Reference* [74].** V_T_ uncorrected for plasma free fraction of radioligand **AD=MCI=HC** V_T_ corrected for plasma free fraction of radioligand **AD > HC**: iP, mTG, iTG, Ent **AD > MCI**: mTG, iTG, Ent SUV_R_ with genotype correction: **AD > HC**: mTG, iTG, iP, Ent, PG **MAB > HAB**: in all diagnostic groups. DVR **AD > HC**: mTG, iTG, iP, Prec, Hipp, Ent, PG **AD > MCI**: mTG, iTG, iP, Ent, PG, Occ Positive correlations between PBR28 (combined mTG-iTG) and CDR	[75]
^11^C-PBR28 No amyloid detection	5 HAB-MCI/AD 9 MAB-MCI/AD 3 HAB-HC 5 MAB-HC	MCI/AD: (65) HC: (62)	MCI/AD: 14–30 (22) HC: >29 (30)		0–90 min V_T_ (n = 17). DVR (n = 17) SUV_R_ with Ref_CE_ (60–90 min) 2nd PBR28 scan sessions: 1.2–5.7 years after.	AD: 10 AchE inhibitor +, 4 AchE inhibitor- at baseline, and 8 AchE inhibitor +, 6 AchE inhibitor- at the 2nd examination. Time to the 2nd PBR28 imaging: **AD** (2.5 years) < **HC** (4 years). Follow-up CDR: 5 AD stable and 9 with increased CDR. Magnitude of SUV_R_ increase **AD > HC**: iTG, mTG, iP, Prec, Occ, Hipp, Ent. Magnitude of DVR increase **AD > HC**: iTG, mTG, iP, Prec, Occ, Ent. Annual increased in PBR binding: **AD with increased CDR > stable AD** Correlation between increase in PBR28 (pF, Prec, supP, iP) and increase in CDR score.	[76]
^18^F-DPA-714 ^11^C-PiB	12 HAB-AD 12 MAB-AD 2 LAB-AD 17 HAB-proAD 17 MAB-proAD 4 LAB-proAD 11 HAB-HC-A^−^ 9 MAB-HC-A^−^ 4 LAB-HC- A^−^ 2 HAB-HC-A^+^ 4 MAB-HC-A^+^ 2 LAB-HC-A^+^	AD: 8/22 (27) proAD: 16/22 (42) HC-A^−^: 6/18 (25) HC-A^+^: 4/4 (50)	AD: (68.3) proAD: (67.8) HC-A^−^: (68.2) HC- A^+^: (74.3)	AD: (15.8) proAD: (24) HC-A^−^: (29.5) HC-A^+^: (29.1)	0–90 min. SUV_R_ with Ref_CE_ (60–90 min) VWA-SPM Clinical follow up: 2 years	SUV_R_ **LAB-AD=LAB-proAD=LAB-HC** **HAB-AD > HAB-HC**: pCin, T, Prec **HAB-proAD > HAB-HC**: GCI, medCin, pCin, P, T, Prec **MAB-AD > MAB-HC**: P, T, Prec **MAB-proAD > MAB-HC**: GCI, P, T, Prec VWA-SPM **HAB/MAD AD > HC**: P, T **HAB/MAD proAD > HC**: F, P, T **At baseline:** **Slow decliners** (stable CDR) > **Fast decliners** (≥0.5 CDR) No correlation between age and DPA in the whole population. Positive correlations between DPA (global cortical binding) and MMSE scores, grey matter volume. In different regions: positive correlations between DPA and PiB	[77]
^18^F-DPA-714 No amyloid detetion	33proAD-19AD: 22 HAB-AD 30 MAB-AD 9 HAB-HC 8 MAB-HC	(dnf)	proAD/AD: (67) HC: (69)	proAD/AD: (21) HC: (29)	0–90 min. SUV_R_ with Ref_CE_ (60–90 min) Clinical follow up: 2 years defining slow and fast CDR and MMSE decliners. Slow: stable CDR or ΔMMSE = −1; Fast: increase in CDR or ΔMMSE = −8	SUV_R_ **proAD = AD** **HAB/MAD AD > HC:** GCI, medCin, pCin, F, P, T, Prec, Occ **At baseline:** **proAD slow decliners** (CDR, n = 11) **> Fast decliners** (n = 22) **proAD slow decliners** (ΔMMSE, n = 15) > **Fast decliners** (n = 18) **AD slow decliners** (CDR, n = 6) > **Fast decliners** (n = 13) **AD slow decliners** (ΔMMSE, n = 8) > **Fast decliners** (n = 8) Positive correlations between DPA (global cortical binding) and MMSE scores	[78]

The TSPO density was evaluated in several studies using different radioligands, population characteristics and methods of analyses. When the rs6971 polymorphism is considered, groups are presented as HAB, MAB and LAB in addition to AD and HC. *Ref* is the reference region. The *Ref_Cluster Analysis_* means that authors used a cluster analysis to extract voxels with normal ligand kinetics to serve as the reference input function. *SUR* and *SUV_R_* correspond to standard uptake ratio and standard uptake value ratio using a reference region. *VWA-SPM* indicates a voxel-wise analysis of parametric images using statistical parametric mapping. Empty spaces in the table correspond to data not clearly given in papers. ***Current abbreviations:*** * PK11195 binding it is not sensitive to the effect of polymorphism, so the evaluation of the rs6971 polymorphism is not informative in the studies with this radiotracer; ** concern the whole population; A^+^, amyloid status positive; A^−^, amyloid status negative; AD, Alzheimer’s disease; CDR, clinical dementia rating; dnf, data not found; DVR, distribution volume ratio; GCI, global cortical index; HAB, high-affinity binders; HC, healthy control; LAB, low-affinity binders; MAB, mix affinity binders; MMSE, Mini-mental state evaluation; pro, prodromal; modAD, moderate AD; V_T_, volume of distribution. List of abbreviations of brain areas: Am, amygdala; aCin, anterior cingulate; aOG, anterior orbital gyrus; bG, basal ganglia; CE, cerebellum; Ent, enthorinal; F, frontal cortex areas; fG, fusiform gyri; fG, fusiform gyri; Hipp, hippocampus; infF, inferior frontal cortex; iPG, inferior parietal gyri; iTG, inferior temporal gyri; Ins, Insula; latF, lateral frontal region; latOcc, lateral occipital; latT, lateral temporal; medOG, medial orbital gyrus; medT, medial temporal; medCin, medium cingulate; MT, mesotemporal region; mFG, middle frontal gyrus; Occ, occipital cortex; oF, orbitofrontal cortex; P, parietal cortex areas; Pall, pallidum; PG, parahippocampal gyrus; postCG, post central gyrus; pCin, posterior cingulate; postT, posterior temporal; preCG, pre central gyrus; Prec, precuneus; pF, prefrontal cortex; Pu, putamen; StG, Straight gyrus; St, Striatum; supFG, Superior frontal; supPG, superior parietal gyrus; supTG, superior temporal gyrus; T, temporal cortex areas; Th, thalamus; WC, whole cortex.

**Table 2 cells-09-01941-t002:** Main results of TSPO PET/SPECT studies in Schizophrenia.

Radiotracer	Population	m/f (%males)	Age Range (mean)	Main Information	TSPO Methodology: Scan Acquisition and Analysis	Main Results	Reference
^11^C-PBR28	8 HAB-FEP 8 MAB-FEP 9 HAB-HC 7 MAB-HC	FEP: 11/5 HC: 7/9	FEP: (28) HC: (26)	Nicotine 2 in FEP 0 in HC FEP: 5 with benzodiazepine treatment	0–91 min V_T_	**FEP < HC**: F, T, Hipp	[106]
^11^C-DPA-713	8 HAB-SCZ 4 MAB-SCZ 2 LAB-SCZ 9 HAB-HC 5 MAB-HC 2 LAB-HC	SCZ: 11/3 (79) HC: 9/7 (56)	SCZ: (24) HC: (25)	Nicotine: 2 in HC 3 in SCZ SCZ: 2 unmedicated (last month) 1 with two antipsychotic 11 with one antipsyxchotic [range: 0–1119 chlorpromazine equivalent]	0–90 min V_T_ Logan	LAB were excluded. **HAB-SCZ = HAB-HC** **MAB-SCZ = MAB-HC**	[107]
^18^F-FEPPA	14 HAB-FEP 5 MAB-FEP 14 HAB-HC 6 MAB-HC	FEP: 12/19 (63) HC: 9/11 (45)	FEP: (28) HC: (28)	FEP: 14 antipsychotic naïve, 5 unmedicated (last 4 weeks)	0–125 min V_T_	**HAB-FEP = HAB-HC** **MAB-FEP = MAB-HC**	[108]
^18^F-FEPPA	10 HAB-SCZ 6 MAB-SCZ 1 LAB-SCZ 19 HAB-HC 8 MAB-HC 0 LAB-HC	SCZ. 10/6 HC: 10/17	SCZ: (43) HC: (44)	SCZ: 16 with antipsychotics and others (antidepressants…)	0–125 min V_T_	**HAB-SCZ = HAB-HC** **MAB-SCZ = MAB-HC**	[110]
^11^C-PK11195	16 SCZ 16 HC	11/5 11/5	(33) (33)	Nicotine 11 in SCZ 0 in HC SCZ: 8 antipsychotic-free	0–60 min BP_SRTM_ with Ref_CE_	Nicotine status SCZ > HC BP_SRTM_ **SCZ antipsychotic-free = HC** SCZ > SCZ antipsychotic-free (83% of increase but *p* = 0.097)	[109]
^11^C-PK11195	19 SCZ 17 HC	16/3 14/3	(24) (47)	Nicotine 13 in SCZ 5 in HC SCZ: 4 antipsychotic-free, 16 with antipsychotic	60 min BP_SRTM_ with Ref_Cluster Analysis_	**SCZ = HC**	[113]
^11^C-PBR28	7 HAB-UHR 7 MAB-UHR 10 HAB-HC 4 MAB-HC 13 HAB-SCZ 1 MAB-SCZ 14 HAB-HC	UHR: 7/7 (50) HC: 4/10 SCZ: 3/12 3/12	UHR: (24) HC: (28) SCZ: (47) (46)	2 UHR with citalopram by the past	0–90 min. DVR (V_T_ target/V_T_ whole brain)	**UHR > HC:** Gm, F, T **UHR excluding 2 subjects with citalopram > HC:** F **HAB-UHR > HAB-HC:** Gm **MAB-UHR > MAB-HC:** Gm **SCZ > HC:** F, T UHR: Positive correlations between the mean PBR28 DVR and CAARMS	[105]

The TSPO density was evaluated in several studies using different radioligands, population characteristics and methods of analyses. *Ref* is the reference region. The *Ref_Cluster Analysis_* means that authors used a cluster analysis to extract voxels with normal ligand kinetics to serve as the reference input function. Empty spaces in the table correspond to data not clearly given in papers. ***Current abbreviations:*** SZC, patients with schizophrenia; DVR, distribution volume ratio; FEP, first episode patients; HAB, high-affinity binders; HC, healthy control; LAB, low-affinity binders; MAB, mix affinity binders; UHR, ultra-high-risk patients; V_T_, volume of distribution. List of brain areas abbreviations: F, frontal cortex areas; Hipp, hippocampus; T, temporal cortex area.

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
