# Peer review of "In Vivo TSPO Signal and Neuroinflammation in Alzheimer’s Disease"

_cells, 2020, doi:10.3390/cells9091941_

Round 1
Reviewer 1 Report
The current review examines the TSPO signal and neuroinflammation in AD. The article is thorough and well written. The title should probably drop Schizophrenia. There is next to nothing about schizophrenia in the article. It is fine to add the information on schizophrenia but I do not believe it should be in the title as this is really a review on TSPO signaling in AD.
Author Response
Response to Reviewer 1 Comments
Point 1. The title should probably drop Schizophrenia. There is next to nothing about schizophrenia in the article. It is fine to add the information on schizophrenia but I do not believe it should be in the title as this is really a review on TSPO signaling in AD.
Response 1: We thank the reviewer for his work. At his request, we changed the title of our article: "In vivo TSPO Signal and Neuroinflammation in Alzheimer's Disease".
Reviewer 2 Report
In this review Tournier et al survey the literature on TSPO and neuroinflammation in vivo. The review is divided into three parts:
The first part will describe the detection of TSPO levels by in vivo or in situ imaging mainly in Alzheimer's disease. The second part will discuss the cellular origin of TSPO while, in the last part, the physiological and pathological role of TSPO is discussed along with its link with amyloid and Tau.
As a general comment the manuscript is very well written, concepts are clear and literature is well covered. The topic is interesting and this contribution is a nice overview on the role pf TSPO in neuroinflammation in vivo.
I found this manuscript very interesting, informative and well written. It certainly deserves to be published as it stand.
Author Response
Response to Reviewer 2 Comments
We thank reviewer 2 for his proofreading work.